# Measuring few-shot extrapolation with program induction

## Abstract

Neural networks are capable of learning complex functions, but still have problems generalizing from few examples and beyond their training distribution. Meta-learning provides a paradigm to train networks to learn from few examples, but it has been shown that its most popular benchmarks require very limited generalization capabilities. Program induction lies at the opposite end of the spectrum: programs are capable of extrapolating from very few examples, but we still do not know how to efficiently search for complex programs. We propose a common benchmark for both communities, measuring extrapolation from few examples coming from the execution of small programs. These are obtained by leveraging a C++ interpreter on codes from programming competitions; extracting small sub-codes with their corresponding input-output pairs. Statistical analysis and preliminary human experiments show the potential of this benchmark for enabling progress in few-shot extrapolation.

## 1. Introduction and motivation

Despite their great successes learning complex functions, neural networks still require large amounts of data and have trouble generalizing beyond their training distribution. In contrast, program induction lies at the opposite end of both spectrums: we can infer programs from few examples that extrapolate far beyond the training samples, but these programs are typically very simple. As a field, we're trying to find the best of both worlds: learning complex functions that generalize broadly from few examples.

Program induction has seen a lot of growing interest, but there is also a lack of large-scale few-shot program induction benchmarks. Most datasets are manually created by a small team of researchers; this creates biases, often assumes a

[1]Anonymous Institution, Anonymous City, Anonymous Region, Anonymous Country. Correspondence to: Anonymous Author <anon.email@domain.com>.

Preliminary work. Under review by NeurIPS 2020 Computer-Assisted Programming Workshop. Do not distribute.

*Figure 1.* Examples of two (relatively simple) tasks coming from programs in our (meta-)dataset. Each task consists of a type signature, 10 training input-output pairs, 10 test input-output pairs and a program that solves them and extrapolates to unseen inputs.

very specific Domain Specific Language (DSL) and limits the number of total tasks to a few hundreds. There is also, to the best of our knowledge, no benchmark that allows testing a broad spectrum of program complexity, from 1-line programs to 100-line implementations requiring algorithmic insights. In this work, we present a benchmark containing such a spectrum of problems, with the hope that it provides a useful target both now and in the coming years.

We propose to create a wide range of tasks, ranging from simple one-line programs to complex algorithmic implementations, by extracting sub-codes from codes online. More concretely, we leverage a database coming from the popular website `codeforces.com`, which has hundreds of problems and hundreds of thousands of implementations. From these, we can extract tens of thousands different sub-codes, each describing its own task. We can obtain input-output examples for each code by running the entire program on a set of inputs and recording the input-output example for each sub-code. C++ is, by far, the most popular language in these competitions. C++ has the advantage of being typed and structured, which, as we will see, allows us to more easily categorize and analyze the difficulty of problems. At the same time, C++ is also compiled, which makes it quite hard to collect the evaluation of sub-codes, since that requires running C++ as an interpreted language. Despite this difficulty, we manage to obtain such evaluations for a broad subset of the codes, which allows us to create a rich, diverse meta-dataset with over 10,000 tasks.

Our main contribution is therefore a few-shot benchmark for the meta-learning and program induction communities with the following key aspects:

1. **The dataset is automated, allowing us to scale to thousands of real-world tasks** and removing some of

the biases of manually created datasets. At the same time, tasks are not random, each coming from a solution to a task similar to those given in tech interviews.

2. **Problems have an accompanying solution in the form of a code**, which can also function as a target for methods that are learning to search in program induction.

3. **Tasks have a diverse range of difficulty and can be classified into many different groups**: from their type signatures, to whether the program that generated them had loops, or its number of lines. This allows us to measure methods that can only work with very short programs or with certain input and output types, as well as complex methods tackling long diverse programs; all under the same framework.

## 2. Related work

Related work on the meta-learning community can be found in appendix A.

**Program induction datasets** There have been a number of program induction benchmarks, such as those used in DreamCoder (1), and FlashFill (2; 3), as well as the Abstract Reasoning Challenge(ARC) (4), a list functions benchmark (5), or the SyGus competition (6). Although these benchmarks contain many interesting problems, they have been manually created by humans instead of being automatically generated from natural data. This creates significant bias on the datasets (often being captured by a relatively simple Domain Specific Language) and restricts the amount of tasks to a few hundreds to a bit more than a thousand tasks. In contrast, our benchmark contains more than 6,000 tasks and we estimate we will be able to extend it to around 100k. Such large datasets have been shown to be useful to learn to search (7); however, in contrast to this work, our programs are not random, and can thus capture the structure of real programs. This will allow neural-based methods, often data-inefficient, to learn to search in these domains with less need to embed biases into the search.

**Datasets leveraging competitive programming codes** We take the programs scraped by (8) and Dr.Repair (9) from `codeforces.com`. However, their goals are significantly different from ours. Whereas we make the tasks easier by considering sub-codes and creating thousands of few-shot learning tasks, they learn to go from line-by-line pseudo-code to code (8) or learn to debug programs (9). Even though the origin of the data is the same, our end-product is orthogonal to theirs. (10) is probably closest to our benchmark, manually describing the function of a bit more than 2000 codes with a problem statement. There are two problems with this approach: first, since it is much easier to describe what the code does than defining a task that the code solves, most statements resemble pseudo-code, which turns the task into something closer to translation. Second, because codes are manually annotated, it is hard to scale to tens of thousands of tasks.

## 3. Description of the dataset

We anonymize the name of our dataset because it contains the name of our institution.

### 3.1. Description of the data

In competitive programming there are different *problems*, each with a short text describing the requirements of the target program. There are also multiple test-cases (some public, some private) that the submitted program has to satisfy. For each *problem* we have many hundreds of *codes* that solve it; meaning we have a pair of (code, test-cases) such that the entire code satisfies the test-cases.

For each code we can obtain *sub-codes*: valid continuous segments of code contained in the original code. To be valid, a sub-code has to be correctly parenthesised: start and end at the same level of indentation and never go to a level above than the starting one in the indentation hierarchy.

Given a sub-code we can generate the data for a task; consisting of 20 input-output pairs (10 training, 10 test). We obtain these by running the entire code and observing the intermediate values at every line. Therefore, a task consists of: (1) a type signature describing the types of inputs and outputs, (2) 20 pairs of input-outputs examples, 10 for training and 10 for test, (3) a code that solves these pairs and extrapolates to other inputs. Note that the input distribution is far from random, as it is affected by previous computations in the overall program. Figure 1 shows the example of two tasks.

### 3.2. Overview of the implementation

In this section we provide an overview of how we obtained the data. This helps provide a better understanding on the data distribution, explaining how we obtain the input-output pairs as well as some limitations of our pipeline(further explained in appendix B), which constrain some of the problems in our dataset. We obtain the original raw codes from SPoC and DrRepair (8; 9), which scraped `codeforces.com`. In their case, they are interested in analyzing the code itself and executing the entire program (which can be done by compiling it as usual). This gives us around 300,000 codes to 700+ programming problems.

To interpret C++ we use the Cling C++ interpreter (11). Cling performs an elaborate incremental just-in-time compilation that keeps modifying the abstract syntax tree before executing the new piece of code. This allows us to execute

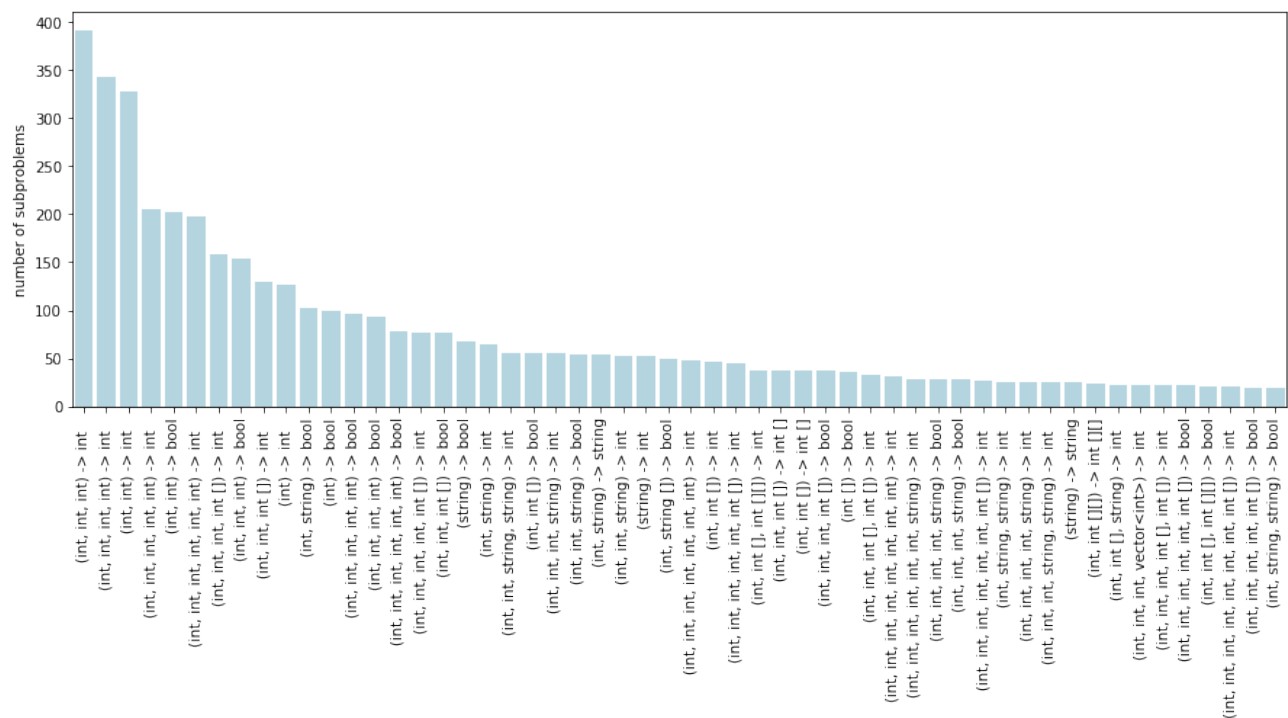

*Figure 2.* Overview of input-output type signatures with at least 20 different sub-problems. Most tasks involve few inputs and output either integers or booleans. Some perform string or array manipulations.

pieces of code and check the values of variables in between. Since they have to be compiled, the given pieces of code have to be self-contained: functions have to be defined entirely before being fed to Cling and loops and if statements have to be given as a block. This severely restricts the type of sub-codes that we can obtain with raw Cling, since we cannot inspect the intermediate values within loops or functions.

We implement a work-around to be able to obtain intermediate values for loops and if statements. First, we standardize all codes changing for loops to while loops plus extra instructions and ensure all loops and if statements are properly bracketed. Once this processing is done, we create an emulator that, instead of feeding the entire while/if statement to Cling, it first calls its condition and then calls the appropriate code depending on whether the condition is satisfied. Note that these if/while conditionals are often very interesting quantities, and we also include them as tasks even though there is no explicit boolean variable created in the original code.

Competitive programming codes interact with the terminal, receiving inputs and outputting results, which Cling cannot handle. We therefore implement a wrapper that simulates this communication. Since console outputs often contain interesting results, we also store them as program outputs. Furthermore, whenever we have an uninitialized variable ("int a;") or a variable initialized within an if statement that wasn't evaluated for a particular test, we mark it as 'null'.

Finally, we often have codes that are implemented differently, but end up producing the same results. Detecting these occurrences is hard to do for arbitrary programs, and often expensive, but we only need to do it once during the creation of the dataset. Moreover, it can also be approximated by checking whether two programs solve the test-cases of one another. We have currently standardized each program by making variable names depend on their order of appearance instead of their original name. Going forward, we plan on removing further symmetries (such as swapping a pair of lines whose order does not affect the output or removing lines that do not affect the final output) by expressing programs as graphs.

### 3.3. Statistical analysis

Figure 2 shows the most popular signatures. As expected, most involve integer manipulations as well as classification problems from few variables. There are other signatures that involve array(list) and string manipulations, often conditioned on other variables like integers or individual characters. These are interesting as they often require to generalize to longer computations as well as bigger data structures. Finally, there are other types that are more complex such as matrices or list of strings.

Figure 3 shows the difficulty of our tasks along three different axis. First, 30% of problems contain either if-statements or loops that require generalizing to up to 10 times more

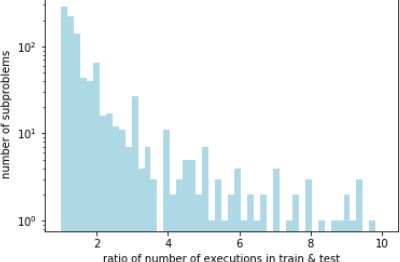 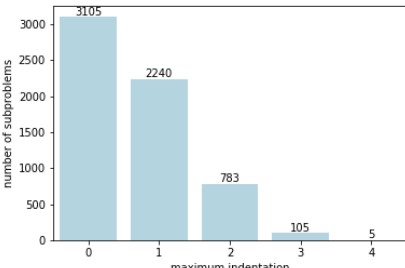 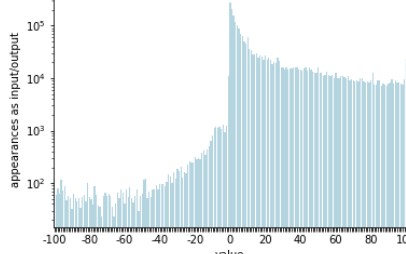

*Figure 3.* (Left) Ratio of number of maximum number of lines executed for a test input vs maximum number of lines executed for a training input; showing the requirement to generalize to longer executions. (Center) Depth of indentation execution (nested loops and if statements), which often significantly affects the difficulty of program induction. (Right) number of times each integer in [-100,100] appears as an input or output on a test-case; note the logarithmic y-axis.

```cpp
bool only_one_solved(int A, int B, int C) {
    B = min(A, C);
    return B % 2 == 0;
}

double no_one_solved(int A, int B, int C) {
    return C/(B * 1.0 / A + 1);
}
```

*Figure 4.* Two examples of difficult problems tested by humans: the first was solved by 1/5 and the second was not solved by anyone. The program on the left of figure 1 was solved by all 5 subjects.

operations than those needed for training examples. Conditional execution (characterized by indentation in C++) is often very hard for program induction techniques. Most programs have a single level of indentation (no conditional execution), but some require multiple up to 4 levels of nested execution. Finally, we observe that most input and outputs involve small positive integers (note the logarithmic y axis), but many involve larger numbers. It is worth noting that these can extend up to $\pm 2 \cdot 10^9$.

## 4. Preliminary human baselines

Humans can infer programs from few examples and extrapolate them beyond the training distribution, but also have a limited search capacity and cannot mentally execute large programs. To assess the difficulty of our dataset, we choose a random subset of 30 problems such that the number of executed lines was at most 5. We tested 5 humans with some prior C++ exposure in high school, but who did not necessarily major in Computer Science.

Subjects saw 10 test-cases and had to describe the program (in natural language) that they believed generated the data. Out of all 30 problems, 13 problems were solved by all 5 subjects, and 10 were solved by some, but not all subjects. Each subject solved between 14 and 19 problems. These

results are encouraging because they show that most problems (at least $25/30 \approx 83\%$) are feasible to infer, with a significant fraction (1/3) being non-trivial. Even for tasks solved by everyone, it is likely that this is still far from what most methods can achieve at the moment, providing a challenging benchmark for the meta-learning and program induction communities.

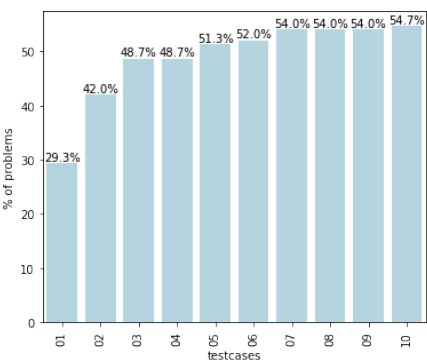

*Figure 5.* Fraction of problems solved by humans after seeing $n$ examples; most problems only require a couple of examples, with significant progress until 5 examples.

## 5. Discussion

We present a new benchmark for few-shot extrapolation and program induction. We hope this sparks progress in making algorithms that (learn to) search complex program spaces from few examples. The structure of this dataset also enables other interesting problems, such as informing the search by surrounding code (since tasks come from subcodes of bigger implementations) as well as text describing the task of the overall code. Finally, it provides a scalable benchmark containing both very simple and very complex problems, all under a single framework to catalyze progress in program induction in the coming years.

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

## A. Extra related work

**Meta-learning**   meta-learning (12; 13; 14) aims at learning priors from many tasks so as to generalize to a new task from few amounts of data; for a nice recent survey see (15). The three main paradigms in meta-learning have been optimization-based approches, MAML being the primary example (16), model-based approaches that tailor to a particular application (often image classification) (17) and architecture-based approaches that use LSTMs, transformers, GNNs, etc to encode the dataset before making a prediction (18; 19; 20). Most of these methods assume that the input form is uniform and do not typically generalize outside broadly outside the data distribution (21), especially non-optimization-based approaches (22). Given that we know tasks are generated by pieces of code, this makes it close to AutoML (23), which often searches through code-like representations to optimize machine learning models.

**Related datasets**   Few-shot learning benchmarks have allowed great progress in meta-learning. The two most popular ones are in few-shot classification for computer vision: miniImageNet (24) and Omniglot (25). Other notable few-shot classification benchmarks have been proposed such as tieredImageNet (26), SlimageNet (27), CUB-200 (28) and meta-dataset (29). There have also been pushes to increase the generality of meta-reinforcement learning benchmarks to include completely different virtual environments (30; 31) as well as learning an entire RL loss functions that generalize between them (32).

## B. Further comments on current limitations and future work

There are a few practical limitation with the implemented pipeline that restrict some codes from being added to our database. This does not affect the correctness of our tasks, but slightly biases the distribution of codes in our benchmark with respect to the distribution of codes in competitive programming as a whole.

In the current version of the dataset (we plan to expand it with even more tasks) we discard codes that contain functions, as Cling cannot analyze them line-by-line. Therefore we cannot obtain sub-codes from pieces of functions. We are currently discarding codes that have functions; however, in the future we will add codes that contain functions, treating them as individual instructions that cannot be split. To restrict the size of the overall dataset as well of the intermediate pipeline, we currently remove test-cases that surpass $10^6$ bits=125KB.

Programs in our tasks consist of contiguous segments of code where the output variable is modified on the last line and input variables are all variables involved in this particular segment. However, this implies that current programs contain lines or variables that do not necessarily affect the input. Understanding these relations requires static analysis and we plan to do it in the near future.