# OpenReview forum: "Measuring few-shot extrapolation with program induction"
_NeurIPS.cc/2020/Workshop/CAP — NeurIPS 2020 CAP Workshop_

### Official Review · AnonReviewer1 · 2020-10-22
**a fairly comprehensive synthesis by example benchmark on executable code fragments**

**Rating:** 7
**Confidence:** 4

**Review:**

quality : pretty well curated dataset, the gradation of difficulty is nice here, as it has a range of easy tasks to hard ones. one  thing that is lacking (correct me if I'm wrong) is that the dataset is "dead" as in it's literally a giant text file without an environment that would allow a researcher to interact with it. does it come with a CFG for the kind of C++ constructs that we would need to be aware of to generate these code fragments? Without an interactive environment to easily generate, execute, and validate these codes on the fly, we would be stuck with more lstm-like generation model, which as shown to be ineffective in synthesis compared to a more execution-guided style of program synthesis.

clarity : good
originality : okay
significance : could be a lot better by making an interactive component, for instance, maybe define an action space in the DSL, such that a random agent can select random actions to generate random code, that would be a great start.

---

### Decision · Program_Chairs · 2020-11-02

**Decision:**

Accept

**Comment:**

The review is positive, so I am recommending acceptance.
I do agree with the reviewer that it would be very cool if there were an environment where it was easy to execute these, etc.
I get the sense that this is being worked toward, though.